# Novel Serum Biomarkers for Patients with Allergic Asthma Phenotype

**DOI:** 10.3390/biomedicines12010232

**Published:** 2024-01-19

**Authors:** Jolita Palacionyte, Andrius Januskevicius, Egle Vasyle, Airidas Rimkunas, Ieva Bajoriuniene, Astra Vitkauskiene, Skaidrius Miliauskas, Kestutis Malakauskas

**Affiliations:** 1Department of Pulmonology, Lithuanian University of Health Sciences, LT-44307 Kaunas, Lithuania; skaidrius.miliauskas@lsmu.lt (S.M.); kestutis.malakauskas@lsmu.lt (K.M.); 2Laboratory of Pulmonology, Department of Pulmonology, Lithuanian University of Health Sciences, LT-44307 Kaunas, Lithuania; andrius.januskevicius@lsmu.lt (A.J.); egle.jurkeviciute@lsmu.lt (E.V.); airidas.rimkunas@lsmu.lt (A.R.); 3Department of Immunology and Allergology, Lithuanian University of Health Sciences, LT-44307 Kaunas, Lithuania; ieva.bajoriuniene@lsmu.lt; 4Department of Laboratory Medicine, Lithuanian University of Health Sciences, LT-44307 Kaunas, Lithuania; astra.vitkauskiene@lsmu.lt

**Keywords:** allergic asthma, biomarker, serum biologically active substance, eosinophil, fraction of exhaled nitric oxide, total immunoglobulin E

## Abstract

In distinguishing the allergic asthma (AA) phenotype, it has been identified that specific biomarkers could assist; however, none of them are considered ideal. This study aimed to analyze three groups of biologically active substances in the serum. Twenty steroid-free AA patients, sensitized to *Dermatophagoides pteronyssinus*, and sixteen healthy subjects (HSs) were enrolled in this study. Blood samples were collected from all patients. Additionally, all AA patients underwent a bronchial allergen challenge (BAC) with *Dermatophagoides pteronyssinus*, all of which were positive, and blood samples were collected again 24 h later. The concentrations of ten biologically active substances were measured in the serum samples, using enzyme-linked immunosorbent assay (ELISA) and the Luminex^®^ 100/200™ System technology for bead-based multiplex and singleplex immunoassays. Descriptive and analytical statistical methods were used. A *p*-value of 0.05 or lower was considered statistically significant. The soluble interleukin 5 receptor subunit alpha (sIL-5Rα) and thioredoxin 1 (TRX1) concentrations were significantly increased, whereas those of tyrosine-protein kinase Met (MET), pentraxin 3 (PTX3), and I C-telopeptide of type I collagen (ICTP) were decreased in the AA group compared with the HS group. A significant positive correlation was noted for sIL-5Rα with fractional exhaled nitric oxide (Fe_NO_), blood eosinophil (EOS) count, and total immunoglobulin E (IgE) levels, and a negative correlation was noted with forced expiratory volume in 1 s (FEV_1_). Moreover, PTX3 showed negative correlations with blood EOS count and total IgE levels, whereas ICTP exhibited a negative correlation with the blood EOS count. In conclusion, this study demonstrated that the serum concentrations of MET, PTX3, TRX1, ICTP, and particularly sIL-5Rα could potentially serve as biomarkers of the AA phenotype.

## 1. Introduction

Asthma is considered to be a chronic inflammatory disorder associated with airway hyper-responsiveness [1]. The recognition of asthma phenotypes is reasonably accurate in relation to gender, age, allergy, family history of allergic disease, commencement of asthma in childhood or adult life, response to inhaled corticosteroid treatment, and others. The most common asthma phenotypes are allergic and non-allergic asthma [2]. Allergic asthma (AA) typically begins in childhood. This particular phenotype is associated with sensitization to aeroallergens. The inhalation of allergens significantly contributes to asthmatic symptoms, leading to acute bronchoconstriction and airway inflammation [3,4]. Therefore, the AA phenotype can be suspected based on the patient’s allergic anamnesis; however, confirmation requires specific immunoglobulin E (IgE) and/or skin prick tests for allergens [4].

Understanding asthma pathophysiology is important for the selection of the best-performing personalized treatment for patients [5]. AA is a type 2 (T2) airway inflammation-driven process [6]; thus, this makes biomarkers of T2 inflammation highly valuable for AA diagnosis and management [7]. Biomarkers associated with the AA phenotype include elevated total IgE levels, sputum and blood eosinophil (EOS) counts, and exhaled nitric oxide (Fe_NO_) [8]. One of the main biomarkers for T2 airway inflammation is the EOS numbers, which are usually analyzed in the blood and the sputum. These EOS counts are the most studied and most widely used biomarker in clinical practice. Increased EOS counts are associated with a higher prevalence of exacerbations and a lower possibility of controlling the disease [9]. Increased total IgE levels are seen in patients with atopic diseases; therefore, they are considered a marker of atopy. IgE has an established role in defining the asthma phenotype, and elevated IgE levels are associated with a more severe course of AA and determine worse symptom control [10]. Total IgE level in blood serum is important in selecting patients with severe AA, for which anti-IgE therapy is indicated [11]. Fe_NO_ is directly measurable in the exhaled breath of patients. It increases proportionally with the intensity of T2 airway inflammation and is linked to eosinophilia [9]. Although these biomarkers provide insights into the pathophysiologic mechanisms underlying AA, none of them are considered ideal and all of them have their flaws [8]. The level of biological markers in the patient can vary depending on the day or hour; moreover, the test results can be influenced by the patient’s weight, exposure to allergens, medications, infection status, and smoking habit [11]. Blood and sputum EOS counts may not be diagnostic but can serve as prognostic biomarkers for treatment responses; serum total IgE levels closely correlate with the risk of AA and the severity of asthma [8]. For these reasons, the study of a single biomarker is not appropriate [12]. It is recommended to test at least two biological markers to increase the sensitivity and specificity of these tests and to better recognize T2 airway inflammation. It is also recommended to repeat tests rather than to rely on a single, one-time test response. The combination and re-examination of several biomarkers may be helpful in patient selection and predicting the efficacy of any treatment targeting T2 airway inflammation [13]. For this reason, it is necessary to identify novel biomarkers of T2 airway inflammation that can better represent allergic inflammation. There are biologically active substances in the serum, which play a role in the pathogenesis of AA, have the potential to serve as biomarkers, can reflect allergic inflammation, and can be used in clinical practice [14]. Based on the literature data, we chose less investigated biologically active substances involved in T2 airway inflammation [15,16,17,18,19]. We divided the substances into three groups via the role the biologically active substances play in AA pathophysiological pathways: circulating inflammatory biomarkers; blood oxidative stress markers; and extracellular matrix proteinases and degradation products. Chronic inflammation-related substances describe those molecules which play key roles in mediating and perpetuating inflammatory responses. We included blood oxidative stress indicators to reflect the cellular redox homeostasis that takes place under chronic airway inflammation conditions. Extracellular matrix proteinases and degradation products exhibit the ongoing extracellular matrix remodeling and collagen turnover associated with chronic inflammation and structural changes in the airways. We chose to investigate ten potential biomarkers for AA to enhance our understanding of allergic inflammation and accelerate the discovery of novel biomarkers.

Thus, this study aimed to analyze biologically active substances in the serum related to chronic inflammation, blood oxidative stress markers, and extracellular matrix factors as potential biomarkers for the AA phenotype and to compare them with validated biomarkers (total IgE levels, blood EOS counts, and Fe_NO_). We performed a bronchial allergen challenge (BAC) using *Dermatophagoides pteronyssinus* to evaluate the value of these substances in allergen-induced inflammatory responses. The evaluation of these bioactive substances may contribute to the discovery of new biomarkers of T2 airway inflammation and targeted treatments for asthma.

## 2. Materials and Methods

### 2.1. Study Design and Population

The participants were recruited from the Department of Pulmonology at the Hospital of the Lithuanian University of Health Sciences (LSMU), from the Kaunas Clinics. Patients with AA were included in the study in two ways. The first was that eligible patients were selected through electronic hospital medical records, and then potentially eligible patients were contacted by telephone. The second was that eligible patients were selected as they arrived in the outpatients’ clinic for an asthma consultation. AA was diagnosed based on asthma symptoms that last more than one year and airway hyperresponsiveness to methacholine or a positive bronchial reversibility test. The healthy subjects (HSs) consisted of healthy volunteers without lung diseases or allergies. The study lasted from the month of October 2020 to the month of December 2022 and included patients with AA and HSs (the inclusion and exclusion criteria are listed in Table 1).

No later than four weeks after the approval of the inclusion and exclusion criteria, all eligible applicants were invited to participate in the study. Prior to initiation of the research, each subject was provided with ample time to review, comprehend, and sign the informed consent. The participants in the HS group visited the hospital twice—for the screening visit and on the experimental day (V1). Meanwhile, the participants in the AA group visited the hospital on three separate days: for the screening visit; then, for V1, the experimental day; and finally, for the experimental day—24 h after V1 (V2). Table 2 shows the research performed at each visit.

Blood samples were collected from all study subjects on V1. AA patients’ blood samples were also re-collected in V2. All collected blood samples were assigned a unique study number. Blood samples were tested for eosinophil count, total IgE, and biologically active substance concentrations. The biomarkers selected for this study are listed in Table 3.

### 2.2. Pulmonary Function Tests

#### 2.2.1. Spirometry

All participants underwent spirometry testing at least three times. For lung function testing, an ultrasonic spirometer was used (Ganshorn Medizin Electronic, Niederlauer, Germany). The performance of this procedure, including details of the techniques utilized, is described in [20].

#### 2.2.2. Bronchial Reversibility Test

The bronchial reversibility test was performed for participants with bronchial obstruction, defined by Z-scores for FEV_1_/FVC < −1.64, in the AA group. This test was performed using a Ganshorn spirometer (Ganshorn Medizin Electronic, Niederlauer, Germany) in which airflow limitations were framed during the screening visit. Beforehand, and 15 min following administration of a salbutamol inhalation (400 mcg), spirometry was performed. In [21], the authors describe the techniques used for performing the procedure.

#### 2.2.3. Methacholine Challenge Test

For the subjects in the AA group, a methacholine challenge test was performed for those who did not show airflow limitations during the screening visit. A dosimeter (ProvoX; Ganshorn Medizin Electronic, Niederlauer, Germany) was used for the inhalation of methacholine. After inhalation, lung function was assessed every two minutes until the completion of the methacholine challenge test once the FEV_1_ had decreased by 20%. The details of the technique for performance of this procedure are described in [20]. 

#### 2.2.4. BAC with *Dermatophagoides pteronyssinus*

The BAC using the *Dermatophagoides pteronyssinus* allergen (DIATER, Madrid, Spain) was performed for all AA group participants. A dosimeter (ProvoX; Ganshorn Medizin Electronic, Niederlauer, Germany) was used for allergen inhalation; thereafter, the BAC with *Dermatophagoides pteronyssinus* ended once FEV_1_ had decreased by 20%. The details for the entirety of this procedure are set out in [20].

#### 2.2.5. Fe_NO_ Test

Fe_NO_ analysis was performed for all study participants. The Fe_NO_ levels were recorded twice for the subjects in the AA group, before and 24 h following the BAC with the *Dermatophagoides pteronyssinus* allergen. Fe_NO_ was measured by a handheld Vivatmo-me device (Bosch Healthcare Solutions, Waiblingen, Germany). The measurements were obtained according to the instructions, and the test was repeated at least two times. Although the device used in the present study is considered easy to use (patients must blow at a specific strength for 10 s into the mouthpiece), three people were not able to successfully measure Fe_NO_.

### 2.3. Skin Prick Testing

A skin prick allergy test with *Dermatophagoides pteronyssinus*, *Dermatophagoides farinae*, dog and cat dander, five mixed grass pollens, birch pollen, mugwort, *Alternaria*, *Aspergillus*, and *Cladosporium* was performed for all study participants. This testing was performed using standardized allergen extracts (Stallergenes, S.A., Antony, France). The technique details for the performance of this procedure are set out in [20].

### 2.4. Blood Tests

#### 2.4.1. Complete Blood Count and Total IgE

All subjects had peripheral blood samples taken and drawn into vacutainers, which contained dipotassium ethylenediaminetetraacetic acid (K2EDTA) (BD Vacutainer^®^; Becton Dickinson UK Ltd., Wokingham, UK). Samples of the routine clinical chemistry assay were directly transported to the hospital laboratory. For the complete blood count test, the XE-5000 (Sysmex, Kobe, Japan) and UniCel^®^ DxH 800 Coulter^®^ Cellular Analysis System automated hematology analyzer (Beckman Coulter, Miami, FL, USA) were utilized. For the total IgE level test, the AIA-2000 automated immunoassay analyzer (Tosoh Bioscience, South San Francisco, CA, USA) was utilized.

#### 2.4.2. Serum Levels of Selected Analytes Measurement

##### Enzyme-Linked Immunosorbent Assay (ELISA)

Five biomarkers—I C-telopeptide of Type I collagen (ICTP), autotaxin (ATX), interleukin-5 receptor α (IL-5Rα), matrix metalloproteinase-10 (MMP-10), and thioredoxin 1 (TRX1)—in serum samples were detected using the ELISA. A lower limit of detection was identified for ICTP—<50 pg/mL (Abbexa Ltd., Cambridge, UK); ATX—<33 pg/mL (Abbexa Ltd., Cambridge, UK); IL-5Rα—<0.15 ng/mL (Invitrogen, Waltham, MA, USA, USA); MMP-10—<3 pg/ mL (Invitrogen by Thermo Fisher Scientific, Waltham, MA, USA); and TRX1—<0.2 ng/mL (Invitrogen, Waltham, MA, USA). Sample collection and the technical details for the performance of these procedures are set out in [22]. For technical reasons, several biologically active substance concentrations were below the sensitivity range; therefore, these results were treated as zero values.

##### Luminex Assay

Luminex^®^ 100/200^™^ System was used to measure the biomarkers’—tyrosine-protein kinase Met (MET), matrix metalloproteinase-7 (MMP-7), matrix metalloproteinase-9 (MMP-9), pentraxin 3 (PTX3), and lipoprotein-associated phospholipase A2 (Lp-PLA2)—concentration in serum samples. First of all, serum samples were added to a mixture of color-coded beads, pre-coated with analyte-specific capture antibodies. Later, biotinylated detection antibodies specific to the analytes of interest were added and formed an antibody–antigen sandwich. The plates were then analyzed using a Luminex^®^ 100 analyzer (Luminex Corporation, Austin, TX, USA). The concentration values were obtained from the mean fluorescent intensity. Standard curves were generated from the reference cytokine gradient concentrations; the concentrations of these cytokines in serum samples were calculated from the standard curves. A lower limit of detection was identified for MET—<14.65 pg/mL (Invitrogen by Thermo Fisher Scientific, Waltham, MA, USA); MMP-7—<5.18 pg/mL (Invitrogen by Thermo Fisher Scientific, Waltham, MA, USA); MMP-9—<0.87 pg/mL (Invitrogen by Thermo Fisher Scientific, Waltham, MA, USA); and PTX3—<42 pg/mL (Invitrogen by Thermo Fisher Scientific, Waltham, MA, USA). For technical reasons, several biologically active substance concentrations were below the sensitivity range; therefore, these results were treated as zero values.

### 2.5. Confirmation of Participation in the Study

All subjects signed the protocol, which was approved for human use by the regional biomedical Ethics Committee (BE-2-58; 19 June 2020). Unique numbers were allocated for all data in the interest of anonymization. Registration of this study with the identification number NCT04542902 was submitted on ClinicalTrial.gov.

### 2.6. Statistical Analysis

Using SPSS statistical software (IBM SPSS Statistics 20; Chicago, IL, USA), statistical analyses were performed. Descriptive and analytical statistical methods were used.

For the assumption of normality in data distribution, the Shapiro–Wilk test was utilized. Since the data distribution did not pass the normality test, we assessed the difference between two independent groups with low numbers of individuals in each group; therefore, we used a nonparametric Mann–Whitney two-sided U-test. We used a nonparametric Wilcoxon matched-pair, signed-rank, two-sided test with an employed pairs test to evaluate the difference between two dependent samples. Spearman’s rank correlation coefficient was used to estimate correlations between two sets of data. The sensitivity and specificity of the indicators were determined by analysis of the receiver operating characteristic (ROC) curve. The statistically significant difference minimal limit for values was *p* < 0.05.

## 3. Results

### 3.1. Study Subject Characteristics

A total of 36 subjects aged from 18 to 49 years old were included in the study (20 with AA and 16 HSs). Table 4 shows the demographic and clinical data descriptive statistics of each group of the study population. The AA and HS groups did not differ significantly in age, sex, or BMI. FEV_1_ (L) also did not differ between the studied groups; however, the difference between the groups was evident when FEV_1_ was expressed as a percentage. In the AA patients’ group blood, EOS counts, Fe_NO_, and total IgE levels were significantly higher than those of the HS group. The FEV_1_ (L) of the AA group decreased 24 h after BAC, while the blood EOS count and Fe_NO_ increased significantly. Meanwhile, total IgE levels did not change significantly after BAC.

### 3.2. Biologically Active Chronic Inflammation-Related Substances as Biomarkers in Asthma

We investigated the serum levels of selected asthma-related proinflammatory substances as potential predictive and prognostic biomarkers. Table 5 provides the concentration data descriptive statistics of each biomarker from each group of the study population.

We found that serum circulating ATX levels in the control group were similar to AA patients before and after BAC (Figure 1A; Table 5). Evaluating the serum concentration of MET, significant differences between the study groups of AA patients and HSs were noted. The MET concentration in the AA group 24 h after the BAC reduced was significantly lower compared to before the BAC (Figure 1B; Table 5). According to ROC curve analysis, the AUC was 0.74, 0.8, and 0.58 (Figure 2A–C). Lp-PLA2 concentration in serum did not differ between healthy controls and both visits of the AA group (Figure 1C; Table 5). Finally, the chronic inflammation-related soluble interleukin 5 receptor subunit alpha (sIL-5Rα) concentration was elevated in AA patients compared to the HS group. Before BAC, circulating sIL-5Rα levels in AA patients remained similar after BAC. These levels were significantly higher compared to the HS group (Figure 1D; Table 5). According to ROC curve analysis, the AUC was 0.88, 0.88, and 0.51 (Figure 3A–C).

### 3.3. Blood Oxidative Stress Indicators as Biomarkers in Asthma

We investigated circulating PTX3 and TRX1 levels to predict oxidative stress in asthmatic patients’ serum and investigate them as a potential biomarker in asthma. A lower concentration of PTX3 was found in the AA group and was significantly different compared to the HS group. The PTX3 concentration in the AA group decreased 24 h after the BAC and was significantly lower compared to the concentration before the BAC (Figure 4A; Table 5). According to the ROC curve analysis, the AUC was 0.89, 0.97, and 0.73 (Figure 5A–C). In the AA group, serum levels of TRX1 were significantly higher than in the HSs. Moreover, concentrations were significantly elevated in the AA group 24 h after the BAC, compared to the results before the BAC (Figure 4B; Table 5). According to the ROC curve analysis, the AUC was 0.7, 0.7, and 0.61 (Figure 6A–C).

### 3.4. MMPs and ICTP Degradation Product as Biomarkers in Asthma

We investigated the serum levels of selected MMPs and ICTP degradation products as potential predictive and prognostic biomarkers in asthma. MMP-7 remained similar between both investigated groups, indicating no potential prognostic value (Figure 7A; Table 5). AA patients were not able to be distinguished from the HS group by the serum MMP-9 concentration. BAC had no effect on MMP-9 concentration in the AA group (Figure 7B; Table 5). The results of MMP-10 did not differ among the asthma groups; serum concentrations of MMP-10 were similar during both visits of the AA patients. No significant differences were observed compared to the HS group (Figure 7C; Table 5). Finally, we investigated the collagen degradation indicating marker, ICTP. Serum levels of ICTP in the AA group were significantly lower compared to the HS group (Figure 7D; Table 5). Moreover, BAC did not affect the ICTP concentration in the AA group (Figure 7D; Table 5). According to ROC curve analysis, the AUC was 0.79, 0.84, and 0.66 (Figure 8A–C).

### 3.5. Biologically Active Substances in Correlation with Validated Asthma Biomarkers, Lung Function, and in Comparison with Each Other

We tested the correlation of validated asthma biomarkers (Fe_NO_, blood EOS count, total IgE) and lung function (FEV_1_) with biologically active substances whose concentrations were different between the researched group and which can indicate their potential prognostic value in asthma. Evaluating the concentration of MET with Fe_NO_, blood EOS count, total IgE, and FEV_1_, no significant correlations were found between the obtained levels (Table 6). A significant positive correlation was observed between sIL-5Rα and Fe_NO_, blood EOS count, and total IgE (Table 6; Figure 9A–C). Moreover, a strong negative correlation between sIL-5Rα and FEV_1_ was obtained (Table 6; Figure 9D). PTX3 had a strong negative correlation with blood EOS count and total IgE (Table 6; Figure 9E,F); however, no significant correlations were found with Fe_NO_ and FEV_1_ (Table 6). No significant correlations were found with PTX3 levels, Fe_NO_, blood EOS count, total IgE, and FEV_1_ (Table 6). ICTP showed a significant negative relationship with blood EOS count (Table 6; Figure 9G); meanwhile, ICTP in correlation with Fe_NO_, total IgE, and FEV_1_ counts were not statistically different (Table 6).

We tested the intercorrelation of those biologically active substances whose concentrations differed between the AA and HS groups. We found that sIL-5Rα had a significant positive correlation with TRX, while PTX3 had a significant negative correlation with TRX1 (Table 7; Figure 10). No other significant correlations were found between different biologically active substance groups (Table 7).

## 4. Discussion

In our study, elevated concentrations of sIL-5Rα and TRX1, alongside decreased levels of MET, PTX3, and ICTP, were identified in the AA group compared to the HSs. Notably, sIL-5Rα showed positive correlations with Fe_NO_, blood EOS count, and total IgE levels. PTX3 exhibited negative correlations with blood EOS count and total IgE levels, while ICTP showed a negative correlation with blood EOS count. Thus, MET, PTX3, TRX1, ICTP, and particularly sIL-5Rα serum concentrations might be associated with the AA phenotype.

Asthma is a heterogeneous disease with various distinct phenotypes. Early recognition of these asthma phenotypes can increase the probability of successful asthma treatment, improve disease outcomes, and prolong survival. Biomarkers or a suite of biomarkers are needed to recognize the AA phenotype as their combined detection can enhance sensitivity. Our study was based on novel circulating biologically active substances involved in T2 airway inflammation, which were divided into three groups: those related to chronic inflammation (ATX, MET, Lp-PLA2, sIL-5Rα), blood oxidative stress (PTX3, TRX1), and MMPs (MMP-7, MMP-9, MMP-10), along with ICTP. All of the researched substances have been linked with asthma; however, our study stands out in several aspects: first, we studied a total of 10 new biomarkers at one time; second, we tested biomarkers in blood serum, a readily available biological medium; third, to our knowledge, these substances have not yet been studied after an allergen challenge.

Asthma-related proinflammatory substances refer to molecules or compounds associated with the inflammatory processes occurring in asthma patients. They play a crucial role in promoting and sustaining airway inflammation in asthma patients. Among these substances, ATX and Lp-PLA2 hold importance in asthma. In vitro studies have demonstrated that ATX activates various cell types implicated in airway inflammation [23]; meanwhile, Lp-PLA2 is an enzyme associated with inflammation and has been implicated in the pathogenesis of various diseases, including asthma [9]. However, our study did not find these proinflammatory substances to have prognostic value.

MET signaling is closely linked to the activation of immune cells and the production of cytokines and chemokines implicated in the pathogenesis of asthma [24]. Experimental studies have demonstrated that MET activation can enhance the recruitment and activation of inflammatory cells in the airway, such as EOS and T lymphocytes. Moreover, MET-induced cytokine release, including interleukin-8 and interleukin-13, can promote airway inflammation and contribute to disease severity [25,26]. A few years ago, a study was conducted to examine MET gene expression in airway biopsies of asthma patients, which confirmed increased MET expression in bronchial epithelial cells and subepithelial inflammatory and resident cells in asthmatic biopsies [27]. In our study, we evaluated serum concentrations of MET as potential biomarkers in AA. Our findings indicated a significant reduction in MET concentration both at baseline and 24 h after the BAC, indicating a potential role for MET in AA and its response to allergen exposure. MET acts as a receptor for hepatocyte growth factor, and when hepatocyte growth factor binds to the MET receptor, it initiates intracellular signaling events that are crucial in asthma [26]. The decrease in MET concentrations in the serum of AA patients might indicate a suppression in the proteolytic cleavage of these receptors, suggesting a more pronounced MET-related signaling and immune cell activation.

The shedding of the extracellular domain of IL-5Rα generates sIL-5Rα, which can be detected in the circulation. Interleukin-5 (IL-5) is a vital cytokine, a signaling molecule that plays a crucial role in regulating the immune system, particularly in allergic and inflammatory responses. It is primarily known for its role in the differentiation, maturation, and activation of EOS, a type of white blood cell involved in immune responses against parasitic infections and allergic reactions [28]. IL-5 binds to its cell surface receptor to initiate signaling cascades that lead to the activation of EOS. Although sIL-5R can still interact with IL-5, it lacks the same signaling capabilities as the cell-bound receptor [29]. Elevated serum concentrations of sIL-5Rα have been noted in asthma patients and in patients with nasal polyposis and have been suggested to play a role in the down-modulation of the eosinophilic inflammatory response [30]. Our findings demonstrate that AA patients exhibit significantly higher concentrations of sIL-5Rα than HSs. Furthermore, these sIL-5Rα concentrations in AA patients are not significantly altered by, but remain elevated after, BAC, suggesting long-term IL-5 signaling and persistent eosinophilic inflammation. The exact mechanisms and factors influencing the shedding and release of sIL-5Rα may vary under different physiological and pathological conditions. sIL-5R serves as a regulatory factor in immune responses involving IL-5. It can bind to IL-5, potentially modulating its effects on EOS activation and inflammation. Measuring sIL-5R concentrations in the bloodstream can offer insights into immune system activity and may hold diagnostic and therapeutic implications in conditions characterized by eosinophilic responses [31]. Moreover, serum concentrations of sIL-5R significantly correlate with Fe_NO_, blood EOS count, total IgE concentrations, and FEV_1_. Combining well-known asthma markers with sIL-5R as a potential biomarker could enhance its predictive value.

PTX3 and TRX1 have been identified as potential markers of oxidative stress in asthma [19,32]. Oxidative stress plays a crucial role in asthma, and these markers have been shown to have associations with oxidative damage and airway inflammation [33]. They provide valuable insights into both systemic and local oxidative stress concentrations in asthma. Measuring these markers in serum provides a noninvasive approach to assess the oxidative stress burden and monitor disease progression. PTX3 is a soluble pattern-recognition receptor that plays a crucial role in the immune response and inflammatory processes [34,35]. In the context of asthma, PTX3 regulates airway inflammation, airway remodeling, and oxidative stress, making it a potential biomarker for asthma severity and a therapeutic target [36]. PTX3 concentrations in sputum have been studied in both pediatric and adult populations with asthma, where it was found to be elevated compared with levels in healthy controls [37,38]. However, investigations into the serum concentrations of PTX3 are currently lacking. Our results demonstrated lower serum concentrations of PTX3 in AA patients, significantly differing from the findings in the HS group. Moreover, the BAC led to a reduction in PTX3 concentrations in AA patients, compared with the levels before BAC. Several factors could explain the decrease in serum PTX3 concentrations in AA patients. PTX3 concentrations in serum can fluctuate during asthma, reflecting the inflammatory processes associated with this respiratory condition. Beyond its role in immunity, PTX3 contributes to tissue repair and regeneration. It helps modulate the inflammatory response, which is crucial for healing damaged tissues. Moreover, we found a significant negative correlation between PTX3 concentration and blood EOS count, indicating a decrease in PTX3 concentrations in highly eosinophilic patients. The immune response in AA is complex, involving activating various immune cells, such as EOS, and releasing inflammatory mediators. As an immune modulator, PTX3 may be influenced by altered immune responses in asthma, with decreased concentrations potentially indicating a shift in the immune profile in favor of other immune molecules or pathways.

TRX1 exerts its antioxidant function by reducing oxidized proteins and maintaining their active state. In the context of asthma, there is accumulating evidence suggesting the involvement of TRX1 in the regulation of airway inflammation and oxidative stress. Studies have shown the altered expression and activity of TRX1 in bronchial epithelial cells, airway smooth muscle cells, and immune cells in asthma patients [39,40,41]. Notably, one study reported that plasma TRX1 concentrations were significantly higher in AA patients than in the study’s HSs [42,43]. This finding suggests that systemic oxidative stress and inflammation in asthma are associated with an increased release of TRX1 into the bloodstream. Moreover, the immune system overreacts to allergens in asthma, leading to inflammation and bronchoconstriction. TRX1 potentially modulates immune cell activities, cytokine production, and the overall immune response in the airways, which can impact the severity and duration of AA symptoms. Our findings highlight the importance of TRX1 in AA. In the AA group, serum TRX1 concentrations were significantly higher than in the HS group, indicating a potential association between TRX1 and AA. This finding suggests that TRX1 might play a role in allergic reactions and could serve as a biomarker for this specific asthma phenotype. Furthermore, TRX1 concentration in the AA group increased significantly 24 h after BAC. This suggests that the allergen exposure triggers a response that leads to the increased production or release of TRX1 into the bloodstream. Such changes in TRX1 concentrations in the serum may indicate a role for oxidative stress and inflammation in asthma, helping clinicians in assessing disease severity and treatment response.

MMPs are a family of enzymes involved in the remodeling of extracellular matrix components, such as collagens and elastin, which play a crucial role in asthma [44]. MMP-7, MMP-9, and MMP-10 are important in asthma, particularly in airway remodeling and extracellular matrix alterations. Although the roles of MMP-7 and MMP-10 in asthma are still being elucidated, MMP-9 has shown consistent associations with disease severity and airway remodeling processes [45]. A key observation in our study was that MMP-7, MMP-9, and MMP-10 concentrations remained similar across all the investigated groups, indicating that they may not possess any prognostic value in asthma.

Additionally, we investigated the collagen degradation marker, ICTP, which showed interesting results. Although ICTP does not directly measure airway remodeling, it serves as a marker for collagen breakdown, which is a key component of the connective tissue in the body, including the airway walls. In theory, increased collagen breakdown contributes to alterations in the airway tissue structure. A previous study examined ICTP concentrations in the sputum of asthma patients and found lower concentrations in these patients than in the HSs in the study [46]. No studies have examined ICTP concentrations in the serum. In our study, the AA group had significantly lower ICTP concentrations than the HS group. In asthma, airway remodeling is characterized by increased production of extracellular matrix proteins [47]. This increased airway tissue mass can result from either heightened extracellular matrix protein production or reduced degradation. The lower ICTP concentrations we observed might explain reduced collagen degradation in the airway tissue. Moreover, ICTP concentrations significantly correlate with blood EOS count—a higher blood EOS count correlates with decreased ICTP concentration. EOSs are a significant source of transforming growth factor β1, which promotes extracellular matrix protein production via airway smooth muscle cells and pulmonary fibroblasts [48]. Combining serum ICTP concentrations and EOS count could offer valuable prognostic information, not only for the AA phenotype, but also for assessing the severity of the disease and degree of airway remodeling.

This study has several potential limitations. First, our investigation was limited to biologically active substances in the serum, and their concentrations in sputum remain unknown. Second, a large-scale study involving a larger number of AA patients and HSs is needed to examine the correlation of AA asthma biomarkers. Third, in this study, we included patients with AA who were allergic to house dust mites, regardless of the presence or absence of allergy to other allergens. Fourth, HSs were not sensitized to allergens, which may have some sensitization effect on T2 biomarkers levels.

## 5. Conclusions

This study highlights the heterogeneity of asthma and the importance of identifying potential asthma biomarkers to enhance treatment effectiveness and outcomes. The identification of novel biomarkers and correlation with validated biomarkers is crucial for accurately diagnosing and characterizing AA phenotypes. The AA group can be better characterized by distinct serum concentrations of MET, sIL-5Rα, PTX3, TRX1, and ICTP as the concentrations of these biologically active substances significantly differ from those in HSs. Moreover, sIL-5Rα displayed significant associations with validated asthma biomarkers, such as Fe_NO_, blood EOS count, and total IgE, in the AA group, indicating its potential role in allergic airway inflammation, thereby making it a potential biomarker. Other biologically active substances, such as PTX3 and ICTP, might also prove valuable in characterizing the AA phenotype. These findings underscore the potential utility of these biomarkers in understanding the molecular underpinnings of AA. Future studies should explore the longitudinal dynamics of these biomarkers, evaluate their specificity and sensitivity in larger cohorts, and assess their potential as prognostic indicators. Additionally, mechanistic investigations into the interplay of these biomarkers in the context of AA pathogenesis may unveil novel therapeutic targets.

## Figures and Tables

**Figure 1 biomedicines-12-00232-f001:**
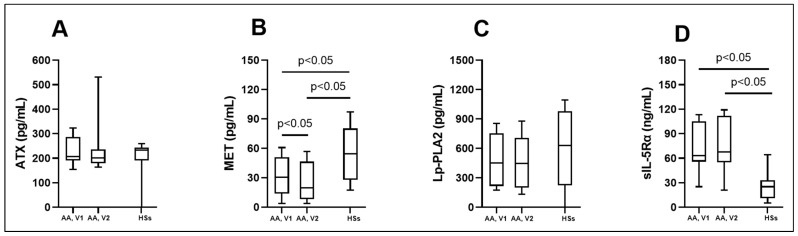
Serum levels of ATX (**A**), MET (**B**), Lp-PLA2 (**C**), and sIL-5Rα (**D**). AA—allergic asthma; ATX—autotaxin; HSs—healthy subjects; Lp-PLA2—lipoprotein-associated phospholipase A2; MET—tyrosine-protein kinase Met; sIL-5Rα—soluble interleukin 5 receptor subunit alpha; V1—experimental day; V2—experimental day—24 h after V1. Data are presented as the median with interquartile range. Statistical analysis: between investigated groups, Mann–Whitney two-sided U-test; within one study group, Wilcoxon matched-pair signed-rank two-sided test.

**Figure 2 biomedicines-12-00232-f002:**
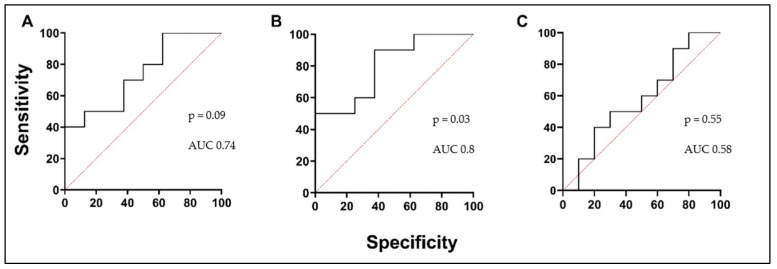
MET ROC curves. (**A**) AA V1 vs. HSs; (**B**) AA V2 vs. HSs; (**C**) AA V1 vs. AA V2. AUC—area under the curve.

**Figure 3 biomedicines-12-00232-f003:**
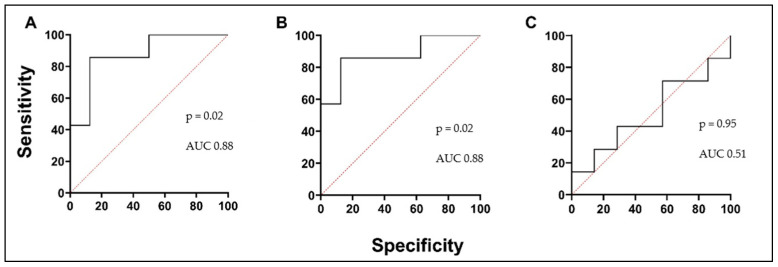
sIL-5Rα ROC curves. (**A**) AA V1 vs. HSs; (**B**) AA V2 vs. HSs; (**C**) AA V1 vs. AA V2. AUC—area under the curve.

**Figure 4 biomedicines-12-00232-f004:**
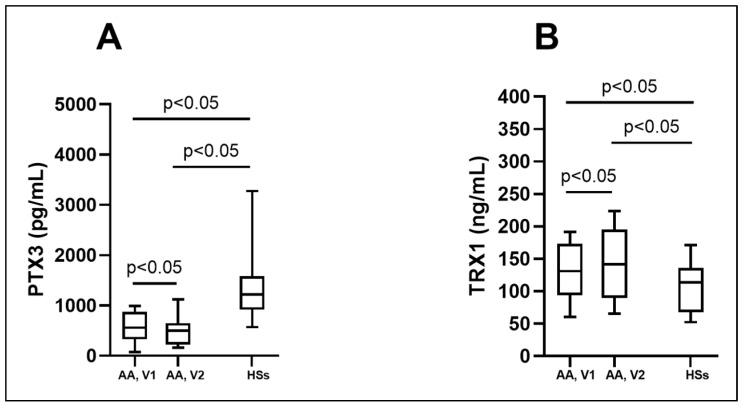
Serum levels of PTX3 (**A**) and TRX1 (**B**). AA—allergic asthma; HSs—healthy subjects; PTX3—pentraxin 3; TRX1—thioredoxin 1; V1—experimental day; V2—experimental day—24 h after V1. Data are presented as the median with interquartile range. Statistical analysis: between investigated groups, Mann–Whitney two-sided U-test; within one study group, Wilcoxon matched-pair signed-rank two-sided test.

**Figure 5 biomedicines-12-00232-f005:**
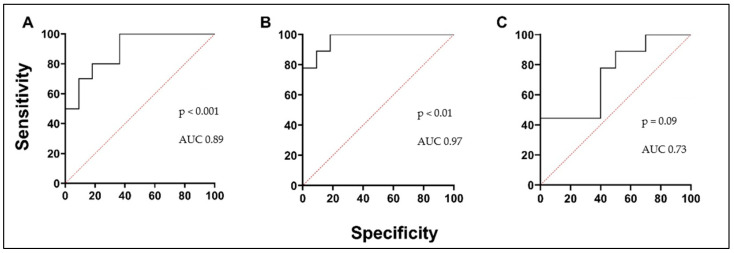
PTX3 ROC curves. (**A**) AA V1 vs. HSs; (**B**) AA V2 vs. HSs; (**C**) AA V1 vs. AA V2. AUC—area under the curve.

**Figure 6 biomedicines-12-00232-f006:**
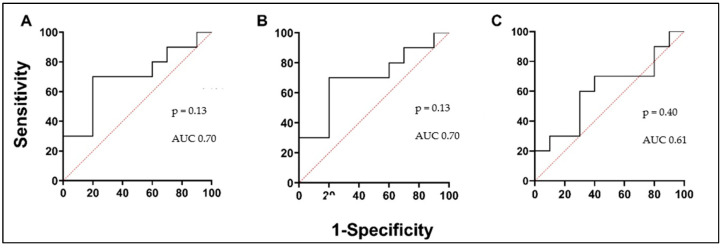
TRX1 ROC curves. (**A**) AA V1 vs. HSs; (**B**) AA V2 vs. HSs; (**C**) AA V1 vs. AA V2. AUC—area under the curve.

**Figure 7 biomedicines-12-00232-f007:**
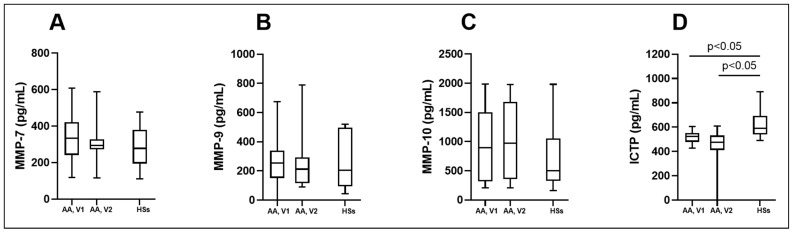
Serum levels of MMP-7 (**A**), MMP-9 (**B**), MMP-10 (**C**), and ICTP (**D**). AA—allergic asthma; HSs—healthy subjects; ICTP—I C-telopeptide of type I collagen; MMP-7—matrix metalloproteinase-7; MMP-9—matrix metalloproteinase-9; MMP-10—matrix metalloproteinase-10; V1—experimental day; V2—experimental day—24 h after V1. Data are presented as the median with interquartile range. Statistical analysis: between investigated groups, Mann–Whitney two-sided U-test; within one study group, Wilcoxon matched-pair signed-rank two-sided test.

**Figure 8 biomedicines-12-00232-f008:**
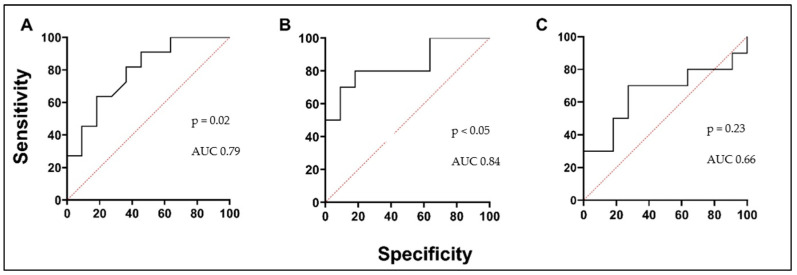
ICTP ROC curves. (**A**) AA V1 vs. HSs; (**B**) AA V2 vs. HSs; (**C**) AA V1 vs. AA V2. AUC—area under the curve.

**Figure 9 biomedicines-12-00232-f009:**
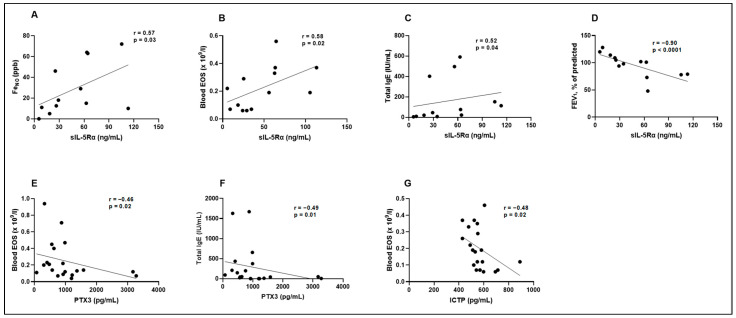
Correlations between sIL-5Rα and Fe_NO_ (**A**), sIL-5Rα and blood EOS (**B**), sIL-5Rα and total IgE (**C**), sIL-5Rα and FEV_1_ (**D**), PTX3 and blood EOS (**E**), PTX3 and total IgE (**F**), ICTP and blood EOS (**G**). EOS—eosinophil; Fe_NO_—fractional exhaled nitric oxide; FEV_1_—forced expiratory volume in 1 s; ICTP—I C-telopeptide of type I collagen; IgE—immunoglobulin E; sIL-5Rα—soluble interleukin 5 receptor subunit alpha; TRX1—thioredoxin 1; PTX3—pentraxin 3; r—correlation coefficient. Statistical analysis: Spearman’s rank correlation.

**Figure 10 biomedicines-12-00232-f010:**
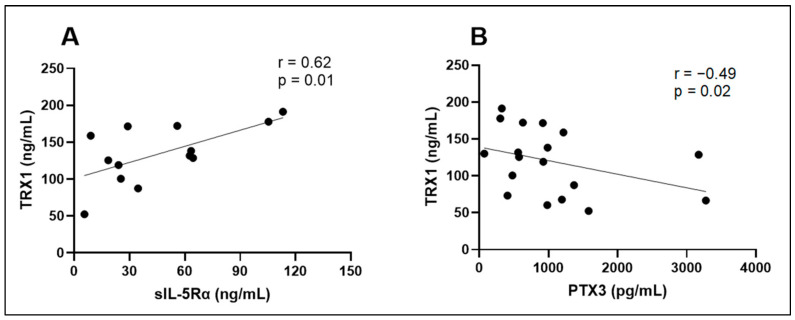
Correlations between sIL-5Rα and TRX1 (**A**), PTX3, and TRX1 (**B**). sIL-5Rα—soluble interleukin 5 receptor subunit alpha; TRX1—thioredoxin 1; PTX3—pentraxin 3; r—correlation coefficient.

**Table 1 biomedicines-12-00232-t001:** Inclusion and exclusion criteria.

Inclusion Criteria		Exclusion Criteria
AA Group	HSs	
Diagnosed with AA Free of inhaled steroids at least 1 month Positive skin prick test for at least *Dermatophagoides pteronyssinus* allergen Positive methacholine challenge or bronchial reversibility test	No symptoms of respiratory disease No clinical allergy symptoms All skin prick tests negative	Age < 18 years Smoker Active airway infection ≤ 1 months prior to study Other clinically significant non-controlled disease

AA—allergic asthma; HSs—healthy subjects.

**Table 2 biomedicines-12-00232-t002:** Investigations at visits.

Screening Visit	V1	V2 *
Inclusion and exclusion criteria	CBC	CBC
Written informed consent	Total IgE	Total IgE
Spirometry	Fe_NO_ measurement	Fe_NO_ measurement
Methacholine challenge test/bronchial reversibility test *	Spirometry	Spirometry
Skin prick testing	BAC with *Dermatophagoides pteronyssinus* allergen *	

BAC—bronchial allergen challenge; CBC—complete blood count; Fe_NO_—fractional exhaled nitric oxide; IgE—immunoglobulin E; V1—experimental day; V2—experimental day—24 h after V1. * only allergic asthma group.

**Table 3 biomedicines-12-00232-t003:** Biologically active substances included in study.

Circulating Inflammatory Biomarkers	Blood Oxidative Stress Markers	Extracellular Matrix Proteinases and Degradation Product
ATX		MMP-7
MET	PTX-3	MMP-9
Lp-PLA2	TRX1	MMP-10
sIL-5Rα		ICTP

ATX—autotaxin; ICTP—I C-telopeptide of type I collagen; Lp-PLA2—lipoprotein-associated phospholipase A2; MET—tyrosine-protein kinase Met; MMP-7—matrix metalloproteinase-7; MMP-9—matrix metalloproteinase-9; MMP-10—matrix metalloproteinase-10; PTX-3—pentraxin 3; sIL-5Rα—soluble interleukin 5 receptor subunit alpha; TRX1—thioredoxin 1.

**Table 4 biomedicines-12-00232-t004:** Demographic and clinical characteristics of the study population.

Characteristic	AA Patients		HSs
Number, n	20		16
Sex, M/F	11/9		6/10
Age, years	22 (19; 35)		27 (25; 30)
BMI, kg/m^2^	22.4 (21.1; 28.7)		22.3 (20.1; 24.2)
Asthma control test, score	21 (12; 23)		NA
PD_20M_, mg	0.27 (0.10; 0.45)		NA
PD_20A_, HEP/mL (range)	10 (1; 10)		NA
	**V1**	**V2**	
FEV_1_, L	3.75 (3.05; 3.94) ^#^	3.41 (2.63; 3.88)	3.66 (3.21; 4.64)
FEV_1_, % of predicted	87 (79; 102) * ^#^	81 (65; 95) *	97 (89; 108)
Fe_NO_, ppb	42 (23; 70) * ^#^	64 (37; 108) *	13 (9; 18)
Blood EOS count, × 10^9^/L	0.31 (0.19; 0.44) * ^#^	0.41 (0.32; 0.57) *	0.10 (0.07; 0.13)
Total IgE, IU/mL	295 (122; 640) *	377 (104; 670) *	13 (4; 44)

AA—allergic asthma; BMI – body mass index; EOS—eosinophil; F—female; Fe_NO_—fractional exhaled nitric oxide; FEV_1_—forced expiratory volume in 1 s; IgE—immunoglobulin E; M—male; NA—not assessed; PD_20A_—the *Dermatophagoides pteronyssinus* allergen provocation dose causing a 20% decrease in FEV_1_; PD_20M_—the provocation dose of methacholine causing a 20% decrease in FEV_1_; V1—experimental day; V2—experimental day—24 h after V1. Data are presented as the median with interquartile range. Statistical analysis: between investigated groups, Mann–Whitney two-sided U-test; within one study group, Wilcoxon matched-pair signed-rank two-sided test. * *p* < comparing with the HS group. ^#^ *p* < 0.05 comparing with the AA group after BAC.

**Table 5 biomedicines-12-00232-t005:** The concentration of biologically active substances in asthma patients and HSs.

Biologically Active Substances	AA Patients		HSs
	V1	V2	
ATX (pg/mL)	206.8 (190.9; 286.3)	201.5 (180.3; 235.5)	233.3 (190.9; 243.9)
MET (pg/mL)	30.5 (14.0; 51.1) * ^#^	19.8 (8.0; 46.6) *	54.3 (28.0; 80.3)
Lp-PLA2 (pg/mL)	451.4 (216.4; 755.1)	445.5 (199.9; 706.6)	629.3 (224.9; 980.8)
sIL-5Rα (ng/mL)	63.4 (55.9; 105.4) *	67.9 (54.9; 112.1) *	25.2 (11.3; 33.2)
PTX3 (pg/mL)	560.1 (326.1; 875.2) * ^#^	498.9 (226.5; 651.1) *	1217.6 (919.8; 1580.1)
TRX1 (ng/mL)	131.2 (93.8; 173.6) * ^#^	142.0 (89.4; 195.6) *	113.9 (67.4; 136.1)
MMP-7 (pg/mL)	332.9 (241.5; 421.3)	295.1 (272.1; 327.7)	278.3 (195.3; 379.2)
MMP-9 (pg/mL)	253.7 (151.4; 339.9)	211.9 (116.9; 293.7)	204.2 (94.8; 498.0)
MMP-10 (pg/mL)	896.1 (318.9; 1501.0)	971.0 (355.7; 1682.6)	502.7 (324.0; 1052.5)
ICTP (pg/mL)	523.0 (478.0; 552.0) *	476.3 (411.6; 531.4) *	589.0 (540.0; 693.0)

AA—allergic asthma; ATX—autotaxin; Fe_NO_—fractional exhaled nitric oxide; HSs—healthy subjects; ICTP—I C-telopeptide of type I collagen; Lp-PLA2—lipoprotein-associated phospholipase A2; MET—tyrosine-protein kinase Met; MMP-7—matrix metalloproteinase-7; MMP-9—matrix metalloproteinase-9; MMP-10—matrix metalloproteinase-10; PTX3—pentraxin 3; sIL-5Rα—soluble interleukin 5 receptor subunit alpha; TRX1—thioredoxin 1; V1—experimental day; V2—experimental day—24 h after V1. Data are presented as the median with interquartile range. Statistical analysis: between investigated groups, Mann–Whitney two-sided U-test; within one study group, Wilcoxon matched-pair signed-rank two-sided test. * *p* < 0.05 comparing with HS group, ^#^
*p* < 0.05 comparing with AA V2.

**Table 6 biomedicines-12-00232-t006:** Correlations between biologically active substance and Fe_NO_, blood EOS count, total IgE, and FEV_1_.

	Fe_NO_ (ppb)	Blood EOS (×10^9^/L)	Total IgE (IU/mL)	FEV_1_, % of Predicted
MET (pg/mL)	r = 0.02 *p* = 0.47	r = −0.07 *p* = 0.39	r = −0.06 *p* = 0.41	r = −0.07 *p* = 0.40
sIL-5Rα (ng/mL)	r = 0.57 *p* = 0.03	r = 0.58 *p* = 0.02	r = 0.52 *p* = 0.04	r = −0.90 *p* < 0.0001
PTX3 (pg/mL)	r = −0.32 *p* = 0.10	r = −0.46 *p* = 0.02	r = −0.49 *p* = 0.01	r = 0.23 *p* = 0.17
TRX1 (ng/mL)	r = 0.25 *p* = 0.18	r = 0.19 *p* = 0.23	r = 0.35 *p* = 0.06	r = −0.23 *p* = 0.18
ICTP (pg/mL)	r = −0.02 *p* = 0.46	r = −0.48 *p* = 0.02	r = −0.11 *p* = 0.32	r = 0.17 *p* = 0.24

EOS—eosinophil; Fe_NO_—fractional exhaled nitric oxide; FEV_1_—forced expiratory volume in 1 s; ICTP—I C-telopeptide of type I collagen; IgE—immunoglobulin; MET—tyrosine-protein kinase Met; PTX3—pentraxin 3, r—correlation coefficient; sIL-5Rα—soluble interleukin 5 receptor subunit alpha; TRX1—thioredoxin 1. Statistical analysis: Spearman’s rank correlation.

**Table 7 biomedicines-12-00232-t007:** Correlations between biologically active substance groups.

	MET (pg/mL)	sIL-5Rα (ng/mL)	PTX3 (pg/mL)	TRX1 (ng/mL)	ICTP (pg/mL)
MET (pg/mL)	–	r = −0.27 *p* = 0.45	r = 0.09 *p* = 0.75	r = −0.35 *p* = 0.22	r = −0.16 *p* = 0.55
sIL-5Rα (ng/mL)	r = −0.27 *p* = 0.45	–	r = −0.38 *p* = 0.20	r = 0.62 *p* = 0.01	r = −0.06 *p* = 0.85
PTX3 (pg/mL)	r = 0.09 *p* = 0.75	r = −0.38 *p* = 0.20	–	r = −0.49 *p* = 0.02	r = 0.31 *p* = 0.19
TRX1 (ng/mL)	r = −0.35 *p* = 0.22	r = 0.62 *p* = 0.01	r = −0.49 *p* = 0.02	–	r = −0.05 *p* = 0.86
ICTP (pg/mL)	r = −0.16 *p* = 0.55	r = −0.06 *p* = 0.85	r = 0.31 *p* = 0.19	r = −0.05 *p* = 0.86	–

ICTP—I C-telopeptide of type I collagen; MET—tyrosine-protein kinase Met; PTX3—pentraxin 3; r—correlation coefficient; sIL-5Rα—soluble interleukin 5 receptor subunit alpha; TRX1—thioredoxin 1. Statistical analysis: Spearman’s rank correlation.

## Data Availability

Data are contained within the article.

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
