# Peer review of "Novel Serum Biomarkers for Patients with Allergic Asthma Phenotype"

_biomedicines, 2024, doi:10.3390/biomedicines12010232_

Round 1

Reviewer 1 Report

Comments and Suggestions for Authors

General comments:

This study by Palacionyte et al aims to investigate the potential role of ten circulating analytes as biomarkers in allergic asthma. Even though the methods appear appropriate, and the data and figures are overall easy to interpret, this manuscript suffers from several unclarities and require significant improvements. Especially I have concerns regarding the selection of subjects, asthma diagnosis and methodology. It is unclear why the participants were chosen based on only one allergen and as I understand from the methods, the authors seem to have excluded measurements of serum analytes that were below the limit of detection, but these should not be excluded since these measurements are not faults, but just biologically low and should be included in the analysis as zero or low arbitrary values. Overall, the manuscript would benefit from a language review. The text is also very lengthy, and I recommend reducing the methods, results and discussion.

Specific comments:

a)      Major comments

1)      The whole manuscript needs improved and clarified language, I would recommend using a professional external evaluator for this. The language is also of very varying quality throughout the manuscript. Especially the methods-section is of poor quality.

2)      It is not clear to me if the patients actually have asthma, inclusion criteria say asthma symptoms for at least 12 months (?), but do these patients have an asthma diagnosis?

3)      From the introduction, it is not clear why there is a need for further biomarkers of AA, Lines 50-63 state that EOS is associated with exacerbations and poorer disease control, and IgE depict the allergic phenotype and severity of allergy, while FeNO is proportionate to the intensity of T2-inflammation. So, what does this established biomarkers not cover about AA that novel biomarkers could cover?

4)      The authors must provide a rational for how the 10 serum biomarkers were chosen for this study and explain the concepts of “chronic inflammation-related”, “blood oxidative stress indicators” and “MMPs and ICTP degradation products” and why these and the analytes they included are relevant for AA?

5)      Why did the authors only include patients with allergy to house dust mite? Did the authors have knowledge of other allergies of the patient (like pollen)? How certain are the authors that the asthma was/is linked to the house dust mite allergy?

6)      He inclusion of controls without asthma and without sensitization creates a problem, are the observed differences between AA and HS attributed to the asthma or the sensitization?

7)      From the methods, it now appears as the authors did perform SPT for other allergens as well, but are patients without Dermatophagoides pteronyssinus-sensitization then excluded from the study, as it appears from the inclusion criteria?

8)      For Table 3, please provide statistics for sex, age and BMI variables.

9)      The labelling for Figures 3-5 is a bit strange, I suggest placing 3-1, 3-2 and 3-3 as one Figure 3 A-J. Same goes for Figures 4 and 5

10)  Since the authors have employed non-parametric testing, Figures and Tables should display medians and range/interquartile ranges, not means and SEs. It is unclear from the manuscript if the box plot figures display means or medians.

11)  Line 139: says “For all AA group participants who were sensitized to D. pteronyssinus”, were not all with AA sensitized to this allergen? Was this not part of the inclusion criteria?

12)  Page 6, section on FeNO test: please shorten this section (compare to other sections) and please also improve the language.

13)  Please clarify the IgE-levels measurements, was it total IgE or allergen-specific IgE, and in that case, which allergens? Please also clarify this in the results section.

14)  Line 183-184 and 200-201: I am very concerned regarding these statements, where complete analytes excluded or only certain values for certain patients? Further, samples below limit of detection should not be removed, they are simply too low to measure (still biologically correct and relevant!) but should then be treated as zero-values or set to an arbitrary value of for example half the detection limit, not excluded. Authors must include all values in the analysis. The sensitivity range depict where a measurable value can be safely assumed, by samples below the sensitivity range are not faults, but low and must be included in the analysis. A low value simply means that the patients have absent or low values of an analyte, and this is as biologically important as having measurable values.

15)  The results 3.2, 3.3 and 3.4 should be shortened and more condensed, the mean levels (should be medians) are already displayed in Table 4, I suggest adding the P-values also in the table and reduced the length of the text in these sections. Example, page 12, lines 319-335 could be summarized as “ we found no differences in the levels of MMP7, MMP9 and MMP10 neither between HS and AA nor before and after BAC among subjects with AA (Figure, Table).

16)  Results, section 3.5, I suggest summarizing these results in a Table displaying r and p-values and reducing Figure 6 to include significant correlations. Also, please add the p-values in the plots of Figure 6.

17)  If not required by the journal, please remove all <0.05 and >0.05 from the results-text, as the level of statistical significance has already been defined in the methods.

18)  The discussion is overall well-written, and I suggest that it could start with a brief description of the main findings.

19)  I think that the discussion should highlight which of the authors findings have not been shown before, what is novel about the presented study? Or have all of these markers already been linked to asthma? Also, the discussion needs more clarification on under which circumstances these substances have been previously connected to asthma, and how and if that differ from measuring them in serum in mild allergic asthma? Further, have any of these substances been investigated after allergen challenge?

20)  The limitations section under Discussion needs depth; “First, study was conducted to explore biomarkers for the AA phenotype” is redundant. It is not a limitation that only one phenotype was investigated but a design choice. Limitations of this study include, for example, defining the AA phenotype based on only sensitization to house dust mite. One further suggestion could be to compare not only AA and HS but HS with atopic comorbidities to find signals specific for AA. This selection of only non-atopic HS is also a limitation of the study.

21)  I suggest shortening the Discussion, it is lengthy as compared to what is presented in the results.

b)     Minor comments

1)      “a pilot study” could be removed from the title, why is this a pilot study?

2)      Figure 1 legend has EOS as an abbreviation but I cannot find eosinophils in Figure 1?

3)      Line 78: please write clarify which are these known biomarkers.

4)      Line 80: please change “their value” to “the value of these substances” to avoid confusion.

5)      Line 84: are the subjects 18-80 years old? They seem to be between 25-30 from Table 3. This is highly relevant as measured biological compounds are strongly influenced by age. Please provide median with full range for the age variable in Table 3, along with comparative statistics (Mann-Whitney U).

6)      Line 96: “Sample time” should read ample time?

7)      Line 97: the sentence “Once all subjects signed the informed consent form, research began”. This sounds unscientific.

8)      Line 97-98: the sentence “After arrival at the hospital, participants were required to stay for 30 minutes to 2 hours”, can be removed.

9)      Line 102: “2 table” should be “Table 2”

10)  Line 109: recollected” should be “re-collected”

11)  Line 111: “where the analysis was performed” – referring to which analysis?

12)  I suggest removing Figure 2, as this is lab standard procedure.

13)  Line 124-135: how was “bronchial obstruction” defined?

14)  Line 171-172: sentence “These blood tests were strictly monitored and controlled during processing. The process for the detailed analysis of blood is depicted below.” could be removed, also the detail depicted below only describe the serum analytes.

15)  Line 188-189: I don’t understand “The biomarkers were researched according to the manufacturer’s instructions”

16)  Lines 203-204: this sentence can be removed.

17)  Line 250: the words “concentrations” seems misplaced.

18)  Line 250: >0.05 should be <0.05 (significant?)

19)  Lines 289 and 290, why are there two p-values but only one comparison?

20)  Line 370: “received” should be “observed”?

21)  Incorrect punctuation on line 24

22)  Line 288: “it as a potential biomarker” should be “them as potential biomarkers”

23)  The use of subsequent is incorrect, e.g., on line 134 and 358. Incorrect verb tense on line 151, incorrect preposition on line 221.

24)  Line 364: you did not test combination, you tested correlations.

25)  Please re-write the following sentence: “We tested the correlations of biologically active substances whose concentrations were different between the researched groups among themselves.” as it is difficult to understand.

26)  Line 558-565, some contributions are in bold font, others are not in bold font.

27)  The data availability statement on line 576 is a circular statement. What does this mean? Data availability means if the data I available to use/inspect by someone else.

Comments on the Quality of English Language

 The whole manuscript needs improved and clarified language, I would recommend using a professional external evaluator for this. The language is also of very varying quality throughout the manuscript. Especially the methods-section is of poor quality.

Reviewer 2 Report

Comments and Suggestions for Authors

·      In general, the manuscript requires a significant editing by a native English speaker.

·      Line 21: Define the study design here.

·      Line 22-23: State the age of participants, the date of sampling took place and the country where the study was conducted.

·      Line 25-26: Define the type of statistical analysis used.

·      Line 35-36: The conclusion does not propose a clear direction for future studies.

·      The introduction is too short and does not provide the gap and the potential contribution of the work (this should be clearly stated at the last paragraph). There are missing references to literature, which are closely related to this work (7 references were used only). The section would benefit from including the following (1) the types of asthma; (2) the diagnostic markers for asthma; (3) the predictive biomarkers for asthma; (4) the differences between asthma and other allergic diseases; (5) the treatment of asthma, including anti-inflammatory drugs, phytochemicals and antioxidants. Please refer to these articles (Phytomedicine. 2024 Jan:122:155149; Int Forum Allergy Rhinol. 2015 Sep:5 Suppl 1:S35-40; Curr Issues Mol Biol. 2023 Jun 10;45(6):5099-5117).

·      Table 1 should be moved to methods.

·      Line 84-86: Include more details on how the participants were recruited from the Department of Pulmonology at the Hospital of Lithuanian University Health Sciences Kaunas Clinics.  

·      Figure 1 should be revised. How many participants were included/excluded?

·      Line 108-143: Data collection should be described in much more details.

·      Line 213-215: Justification of these analyses used is needed. Why Mann–Whitney two-sided U-test and Wilcoxon matched-pair were used? Spearman’s rank correlation coefficient was used to estimate correlations between......

·      Table 3 & 4: Please include a column showing a non-significant and significant P-value (0.05, 0.01).

·      It is unclear the type of analysis used in each table.

·      Line 546-556: The scope for future studies is not clearly mentioned and discussed.

·      Include a list of abbreviations at the end.

Comments on the Quality of English Language

Extensive editing required.

Round 2

Reviewer 1 Report

Comments and Suggestions for Authors

General comments:

With this revision, the authors have significantly improved the manuscript. I still have additional comments and suggestions related to the author responses.

Specific comments:

a)      Major comments

1)      Previous major comment 2: It is still not clear to me how the AA group was selected. For those selected, both through electronic hospital medical records and those selected at the outpatients clinic, was an asthma diagnosis required to be included in the study? How was “non-severe course of AA” defined? For the HS, what is the difference between “No clinical allergy symptoms” and “No atopic dermatitis, allergic rhinitis”

2)      Previous major comment 4: Please provide references for statement on line 85 ending with “involved in the pathogenesis of AA”, I still cannot find any references in the introduction supporting the choice of substances to measure.

3)      Previous major comment 6: The authors have probably not understood the question, if the AA-subjects have asthma and allergy to Dp, while the HS do not have asthma and do not have allergy to Dp, how can the authors know that observed differences in substance levels between AA and HS are attributed to the asthma and not to the Dp-allergy? I suggest that this I mentioned as a limitation.

4)      Previous major comment 10: Please proved the range/interquartile range as a continuum (value x to value y). For example, variable Age would be median (range/interquartile range) hence 22 (lowest/25 percentile age – highest/75 percentile age) for the AA group.

b)     Minor comments

1)      Line 84, I don’t understand the word “disease progression”, this has not been studied in this paper.

2)      Line 85, “research” should be “substances/analytes” to avoid confusion.

3)      Line 96, are biomarkers presented in sub-section 2.1?

4)      Previous major comment 13: thank you for clarifying this, why did the authors choose to measure total IgE, and not Dp-specific IgE? What is the relevance of the total IgE for this study?

5)      For the first part of the added sentences on lines 434-438, one could improve the message of novelty by instead writing: “All of the investigated substances have been previously linked to asthma, however our study stands out in several aspects: first…”

Comments on the Quality of English Language

The language has been improved, I still consider the language to be sub-optimal at some places in the text, but I leave to the editors to decide if the language quality is sufficient

Reviewer 2 Report

Comments and Suggestions for Authors

No further comments.

Author Response

The reviewer had no further comments.